DISCOVERY REPORT

# DnaJ mediates phage sensing by the bacterial NLR-related protein bNACHT25

Amy N. Conte[1], Madison E. Ruchel[1,2], Samantha M. Ridgeway[1], Emily M. Kibby[1], Toni A. Nagy[1], Aaron T. Whiteley[1]*

1 Department of Biochemistry, University of Colorado Boulder, Boulder, Colorado, United States of America, 2 Department of Biology, Front Range Community College, Longmont, Colorado, United States of America

* aaron.whiteley@colorado.edu

The Editors encourage authors to publish research updates to this article type. Please follow the link in the citation below to view any related articles.

## Abstract

Bacteria encode a wide range of antiphage systems and a subset of these proteins are homologous to components of the human innate immune system. Mammalian nucleotide-binding and leucine-rich repeat containing proteins (NLRs) and bacterial NLR-related proteins use a central NACHT domain to link detection of infection with initiation of an antimicrobial response. Bacterial NACHT proteins provide defense against both DNA and RNA phages. Here we investigate the mechanism of phage detection by the bacterial NLR-related protein bNACHT25 in *E. coli*. bNACHT25 was specifically activated by *Emesvirus* ssRNA phages and analysis of MS2 phage escaper mutants that evaded detection revealed a critical role for Coat Protein (CP). A genetic assay showed CP was sufficient to activate bNACHT25 but the two proteins did not directly interact. Instead, we found bNACHT25 requires the host chaperone DnaJ to detect CP and protect against phage. Our data support a model in which bNACHT25 detects a wide range of phages using an indirect mechanism that may involve guarding a host cell process rather than binding a specific phage-derived molecule.

## Introduction

The coevolution of bacteria and their viruses, bacteriophages (phages), has led bacteria to evolve diverse antiphage systems that halt infection. These systems can be a single gene or an operon of genes whose products cooperate to sense phage, amplify that signal, and activate an effector response that is antiviral [1,2]. Antiphage systems are distributed in the pangenome for each bacterial species and any one bacterial strain has a subset of these systems, which tend to colocalize and move between bacteria within mobile genetic elements [3–5]. The collection of antiphage systems within a given bacterium is referred to as the bacterial immune system [1,2].

**Data availability statement:** All relevant data are within the paper and its Supporting Information files.

**Funding:** This work was funded by the National Institutes of Health through the NIH Director's New Innovator Award DP2AT012346 (A.T.W); the PEW Charitable Trust Biomedical Scholars Award (A.T.W.), and a Mallinckrodt Foundation Grant (A.T.W.). M.E.R. is supported in part by the National Institutes of Health through the Bridges to Baccalaureate Research Training Program (T34 GM142601) and Howard Hughes Medical Institute (HHMI) Inclusive Excellence 3 Grant (GT16065). S.M.R. was supported in part by the Undergraduate Research Opportunities Program Individual Grant funded by CU Boulder. E.M.K. was supported in part by the NIH T32 Signaling and Cellular Regulation training grant (T32 GM008759 and T32 GM142607). The funders did not play any role in the study design, data collection and analysis, decision to publish, or preparation of the manuscript.

**Competing interests:** The authors have declared that no competing interests exist.

**Abbreviations:** CP, coat protein; iBAQ, intensity-based absolyute quantification; MOI, multiplicity of infection; MP, maturation protein; NCH1, NACHT C-terminal helical domain 1; PAMPs, pathogen-associated molecular patterns; PFU, plaque forming units; PRRs, pattern recognition receptors; WCL, whole cell lysate.

Bacterial antiphage systems, like all immune pathways, can be categorized as adaptive or innate. Adaptive immune systems, such as CRISPR-Cas, are targeted to specific pathogens and specialize as a result of previous exposure or "immunization". Innate immune systems, on the other hand, target a wide range of pathogens in their native form by detecting conserved features. In this way, innate immune pathways act as the first line of defense.

One component of the innate immune systems of humans, plants, and bacteria are nucleotide-binding domain and leucine-rich repeat containing proteins (NLRs) [6–9]. These proteins are grouped based on their gross similarity to each other, i.e., inclusion of a nucleotide binding domain and leucine-rich repeats, however, their evolutionary histories and relatedness are complicated. The nucleotide binding domains of all NLRs belong to the STAND NTPase family, which can be further subdivided into related but distinct sister clades [10]. Human NLRs (such as NLRC4, NLRP3, and other components of inflammasomes) use NACHT modules [11]. Plant NLRs (such as ZAR1 and other R-proteins that form the resistosome) use NB-ARC modules, a subclass of AP-ATPases [10,12]. Previously, we found that bacteria encode NACHT modules in open reading frames with leucine-rich repeats, these are bona fide bacterial NLRs [9]. In addition, bacteria encode many NACHT modules in proteins with other domains in place of leucine-rich repeats. These proteins are "NLR-related" and we used NACHT modules to reconstruct the evolutionary history of how NACHT modules originated in bacteria and were horizontally acquired by eukaryotes [9]. More broadly, bacteria also encode proteins with AP-ATPase modules and other STAND NTPases. Multiple clades of antiviral ATPases/NTPases of the STAND superfamily (AVAST) systems have been described [10,13,14]. In humans, there are also antiviral STAND NTPases that are not NLRs, such as SAMD9 [15,16].

Bacterial NACHT module-containing (bNACHT) proteins, as with all STAND NTPases, have a tripartite domain architecture [9,10]. The C-terminus is a sensing domain, the central NACHT module enables oligomerization and signal transduction, and the N-terminus is an effector domain. While the effector domain function can often be predicted bioinformatically [17] (e.g., identification of a nuclease domain suggests the antiphage system targets phage/host DNA), the stimuli that activates bNACHT proteins is challenging to discover.

We investigated the mechanism of phage detection by bNACHT proteins. Specifically, we focused on sensing of RNA phage because bNACHT proteins are some of the only known bacterial innate immune pathways identified that are capable of protecting against RNA phage. We found that bNACHT25 is indirectly activated by the coat protein (CP) of the model ssRNA phage MS2, which suggests that host cell processes or components are required for sensing. Our analysis led us to discover that activation of bNACHT25 by phage or CP requires the host chaperone DnaJ.

## Results

### Breadth of RNA phage defense by bNACHT proteins

We previously reported that four bacterial NLR-related proteins provided robust phage defense against both DNA and RNA phages [9]: bNACHT02, 12, 25, and 32.

Each encode a central NACHT module. bNACHT02 is 39% identical to bNACHT12, and bNACHT25 is 81% identical to bNACHT32. bNACHT02 and bNACHT12 fall within clade 14 of bNACHT proteins, which all contain short C-terminal NACHT-associated (SNaCT) domains but lack readily discernible effector domains [9] (Fig 1A). bNACHT25 and bNACHT32 are from clade 6, contain NACHT C-terminal helical domain 1 (NCH1) domains, and N-terminal PD-(D/E) XK family DNase effector domains [9,18] (Fig 1A). Nuclease activity of bNACHT25 and bNACHT32 is required for phage protection, as mutating a conserved active site residue of the endonuclease domain (D48A) abrogated defense (Fig 1B).

To investigate the breadth of RNA phages restricted by NLR-related proteins, we challenged *E. coli* MG1655 express-ing GFP as a negative control or bNACHT proteins with diverse ssRNA phages from the *Fiersviridae* family (Fig 1C and 1D). All four systems defended against MS2 and MS2-like phages (the *Emesvirus* genus), however, none of these sys-tems were capable of defending against Qβ or Qβ-like phages (the *Qubevirus* genus). While MS2 and Qβ have similar genome organizations and infectious cycles [19], they possess little shared sequence identity (38% identical). Within the *Emesvirus* and *Qubevirus* genera, the tested phages are highly related (≥92% identical) (S1 Fig) [20].

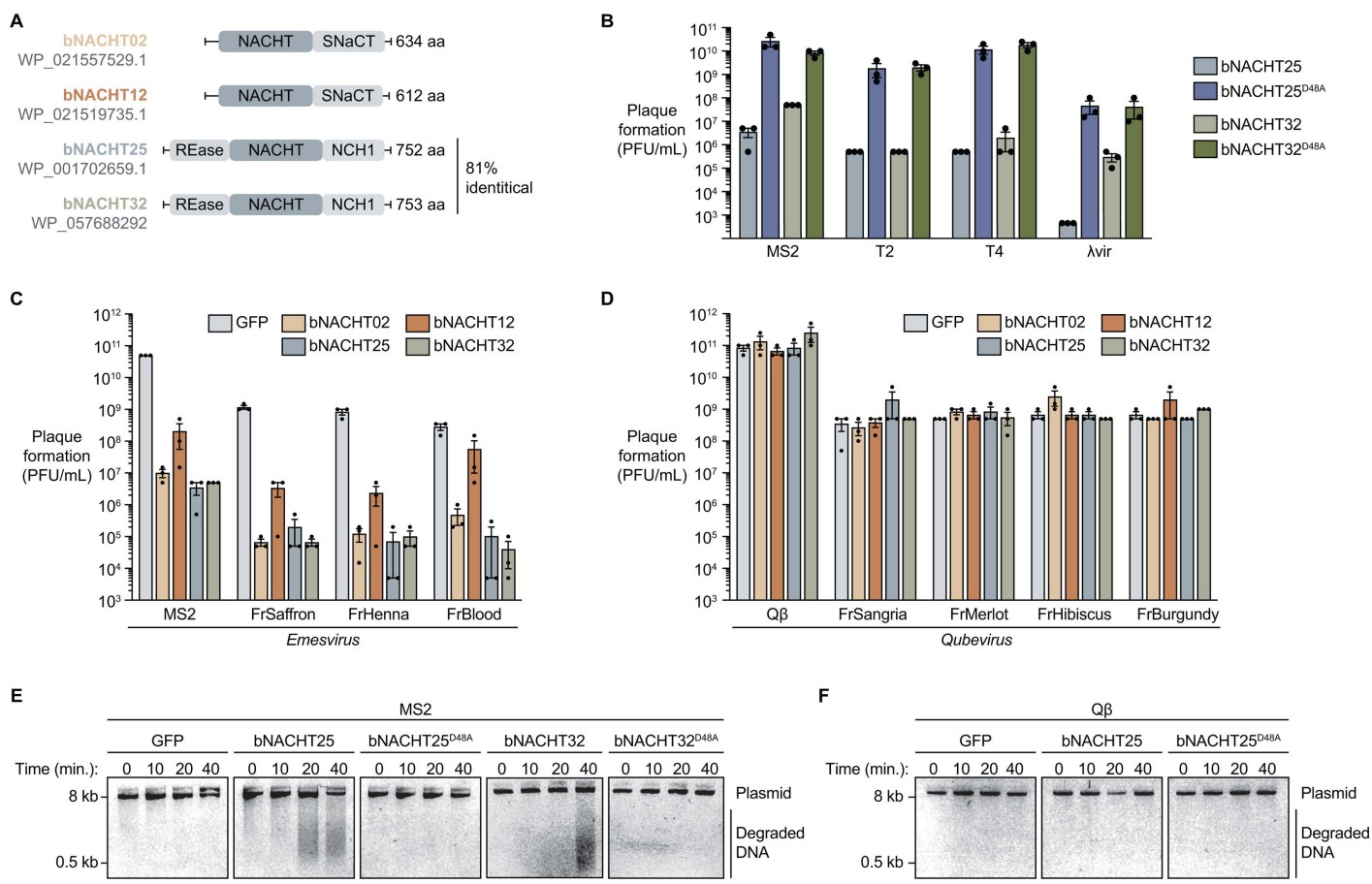

**Fig 1. bNACHT proteins protect against ssRNA phages of the *Emesvirus* genus. (A)** Domain architectures of the bNACHT proteins that pro-tect against MS2. **(B)** Efficiency of plating in plaque forming units per milliliter (PFU/mL) of the indicated phage on *E. coli* expressing bNACHT25 or bNACHT32 with or without the D48A mutation. Data are the mean ± standard error of the mean (SEM) of $n$ = 3 biological replicates. **(C–D)** Efficiency of plating in PFU/mL of the indicated *Emesvirus* or *Qubevirus* ssRNA phages on *E. coli* expressing the indicated defense system. Data plotted as in Fig 1B. **(E–F)** Visualization of plasmid integrity in *E. coli* expressing GFP, bNACHT25, bNACHT25D48A, bNACHT32, or bNACHT32D48A infected with MS2 or Qβ at a multiplicity of infection (MOI) of 2. Plasmid DNA was harvested at the indicated timepoints post-infection and analyzed on an agarose gel. Data are representative images of $n$ = 3 biological replicates. The data underlying this figure can be found in S1 Data.

We confirmed our findings in liquid culture by infecting MG1655 expressing EV or bNACHT25 with either MS2 or Qβ. At a multiplicity of infection (MOI) of 0.2, bacteria expressing EV were lysed by MS2 and Qβ, while cultures expressing bNACHT25 continued to grow after infection with MS2 (S2A–2D Fig). At a MOI of 2, the $OD_{600}$ of bNACHT25-expressing bacteria stopped increasing upon MS2 infection (S2C Fig).

We hypothesized that $OD_{600}$ plateaued when bNACHT25-expressing bacteria were infected due to activation of the endonuclease effector domain and tested this by measuring intracellular DNA degradation following infection with MS2. Plasmid DNA was harvested to represent total host DNA. Infection of bacteria expressing either bNACHT25 or bNACHT32 with MS2 caused DNA degradation indicative of effector activation (Fig 1E), and nuclease activity of both proteins was ablated by introduction of the D48A mutation. Nuclease activity was not observed when cultures were infected with Qβ (Fig 1F); of note, we cannot distinguish whether Qβ evades detection by bNACHT25 or instead encodes a bNACHT25 inhibitor.

## Mutations in CP enable MS2 to evade bNACHT proteins

ssRNA phages have some of the smallest known viral genomes, with MS2 only encoding four genes in its 3.6 kb genome: mat encodes maturation protein (MP), which enables attachment to the F pilus and RNA entry into the host; cp encodes coat protein (CP), which forms the viral capsid; lys encodes lysis protein (L), which allows for host lysis; and rep encodes replicase (Rep), which is an RNA-dependent RNA polymerase that replicates the ssRNA viral genome [21,22] (Fig 2A). We sought to determine how MS2 activated bNACHT proteins and generated spontaneous MS2 escaper mutants that evaded these systems. We isolated 28 MS2 escaper mutants that successfully formed plaques in the presence of either bNACHT02, bNACHT12, or bNACHT32 from three separate, clonal parent lineages. We were unable to isolate MS2 mutants that could grow on bNACHT25. Mutant phage genomes were sequenced and variations from parent genomes were determined (Tables 1 and S1).

Interestingly, MS2 mutants generated on bNACHT32 from two different parent lineages had acquired the same missense mutation in the cp gene, resulting in a G17D mutation (Table 1). Escaper mutant phages encoding CP^G17D evaded bNACHT32-mediated defense compared to their parent strains (Fig 2B and 2C). CP is the major structural protein that forms the viral capsid of MS2 and has been long-studied for its role in RNA binding [23]. The G17D mutation is located on the AB loop of CP, a surface-exposed loop that is not predicted to interact with the RNA genome or participate in the CP dimerization interface [24,25]. Additional mutations in cp were also found in MS2 escaper mutants generated on bNACHT02 and bNACHT12 (S1 Table).

bNACHT32 and the highly similar bNACHT25 were selected for further investigation due to their readily assayable DNase effector domains, which provide unambiguous evidence of activation during phage infection (Fig 1E). We hypothesized that CP activated bNACHT32, leading to effector activation and cell death, and that the G17D mutation in CP reduced the ability of CP to activate the defense system, allowing for MS2 escape. We tested this by inducing expression of CP in the presence of bNACHT32 and measuring colony formation. Significant growth inhibition was observed in the bNACHT32 and CP coexpression condition that was not observed when either was expressed individually (Fig 2D) or when CP was coexpressed with the nuclease-dead bNACHT32^D48A. As suggested by our escaper analysis, CP^G17D did not reduce colony formation to the same degree as wild-type CP when coexpressed with bNACHT32 (Fig 2D).

DNA was purified from cultures expressing CP, bNACHT32, or both proteins combined to visualize effector domain activation. DNA degradation was only observed in the bNACHT32 and CP coexpression condition (Fig 2E), indicating that CP is sufficient to activate the endonuclease domain of bNACHT32 in this genetic assay. The CP^G17D mutant activated bNACHT32 to a lesser extent than wild-type CP (Fig 2E). The decrease in activation was not due to decreased protein expression as CP^G17D was produced in higher amounts than wild-type (Fig 2F). Together, these data suggest that CP plays a role in bNACHT32 activation and the G17D mutations in CP allow MS2 to evade detection.

## CP synthesis is sufficient to activate bNACHT25

We next interrogated the qualities of CP that were required for bNACHT activation using a genetic assay. bNACHT25 was used in place of bNACHT32 (these proteins share 81% identity and >96% similarity) for these and future assays because bNACHT25 provided more robust effector activation in response to phage (Fig 1E). Expression of wild-type CP in the presence of bNACHT25 resulted in inhibition of colony formation and DNA degradation (Fig 2G and 2H). Expression of CP$^{W83R}$, which is unable to oligomerize beyond dimerization and limits capsid assembly [26], similarly induced cell death and DNA degradation, albeit at a slightly slower rate (Fig 2G and 2H). However, mutation of the cp start codon completely

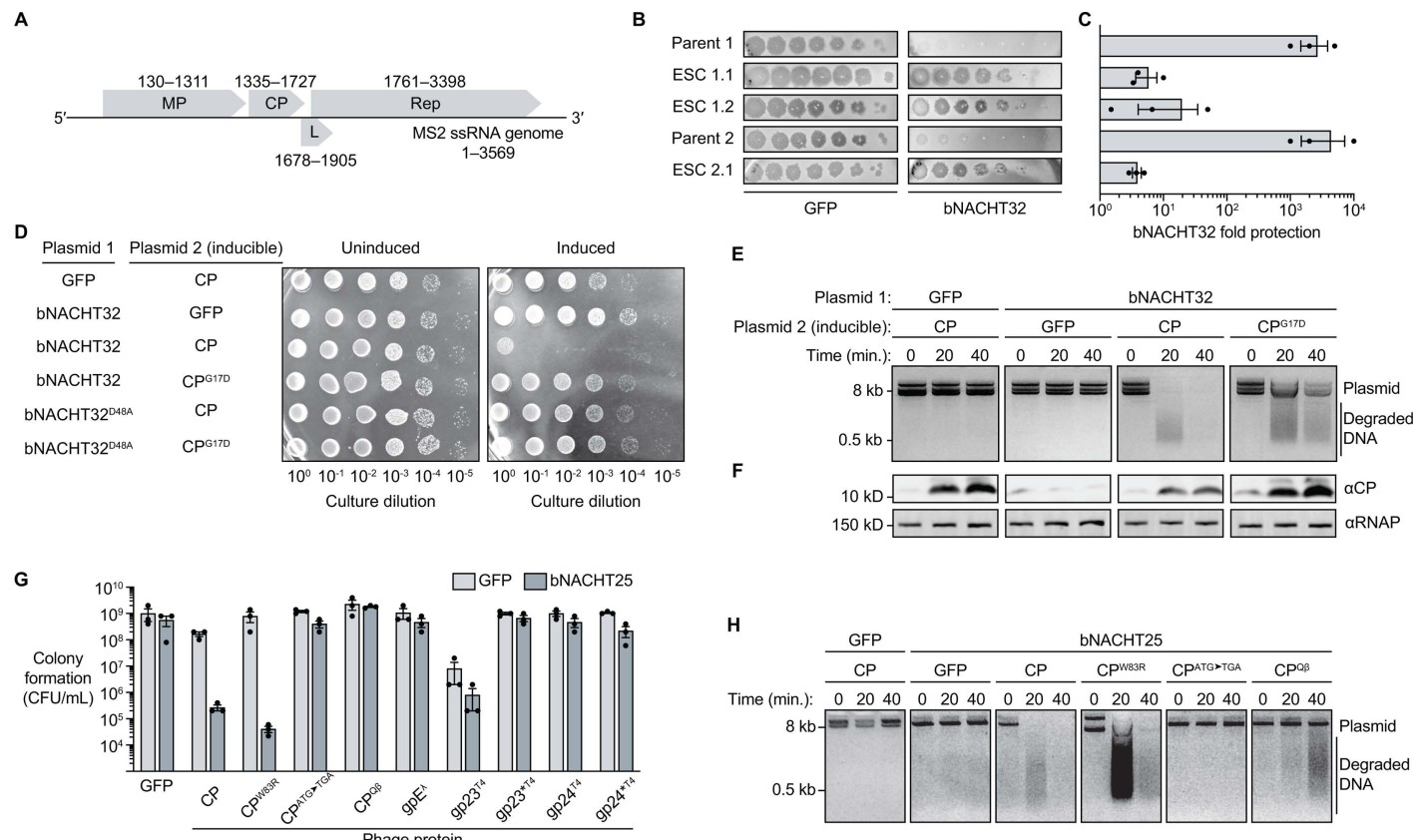

**Fig 2. MS2 phage escaper mutants reveal a role for CP in bNACHT activation. (A)** ssRNA genome of phage MS2. Each gene is labeled with the protein it encodes and the nucleotide positions of the coding sequences. Maturation protein (MP), coat protein (CP), lysis protein **(L)**, replicase (Rep). **(B)** Efficiency of plating of MS2 parent phages and corresponding escaper mutant phages (Escaper, ESC X.Y, where X indicates the parent phage number and Y indicates the escaper phage number) on E. coli expressing GFP or bNACHT32 from a plasmid. Images are representative of n = 3 biological replicates. For ease of comparison, first spot of the dilution series is a 10$^{-1}$ dilution of lysate for parent phages and undiluted lysate for escapers. **(C)** Quantification of data in **(B)** presented as fold protection (PFU/mL of phage plated on GFP divided by PFU/mL of phage plated on defense system). Data are the mean ± standard error of the mean (SEM) of n = 3 biological replicates. See Tables 1 and S1 for bNACHT32 escaper mutations. **(D)** Colony formation of E. coli expressing the indicated plasmids with or without IPTG induction of plasmid 2. Data are representative images of n = 3 biological replicates. **(E)** Visualization of plasmid integrity in E. coli expressing the indicated proteins. Plasmid DNA was harvested at indicated timepoints post-induction of CP or GFP with IPTG. Data are representative images of n = 3 biological replicates. **(F)** Western blot analysis of E. coli lysates from the genotypes, timepoints, and conditions indicated in **(E)**. **(G)** Colony formation of E. coli expressing GFP or VSV-G-bNACHT25 on the chromosome and indicated phage protein from an inducible plasmid. gp23* and gp24* indicate the truncated versions of gp23 and gp24, respectively, which are produced upon proteolytic cleavage during T4 infection [30]. Data are the mean ± SEM of n = 3 biological replicates. **(H)** Visualization of plasmid integrity in E. coli expressing GFP or VSV-G-bNACHT25 from the chromosome and the indicated CP alleles from an inducible plasmid. Plasmid DNA was harvested at indicated timepoints post-induction with IPTG. Data are representative images of n = 3 biological replicates. The data underlying this figure can be found in S1 Data.

**Table 1. MS2 escaper mutants capable of evading bNACHT32.**

| Phage name | NTD (nt 1–130) | mat (nt 130–1,311) | cp (nt 1,335–1,727) | lys (nt 1,678–1905) | rep (nt 1,761–3,398) | CTD (nt 3398–end) |
|---|---|---|---|---|---|---|
| ESC 1.1 | | g162a Q233H L278P | G17D | | | |
| ESC 1.2 | | a69u | G17D | L44P† | u48c† | |
| ESC 2.1 | | a564c | G17D | | | |

Polymorphisms detected in MS2 mutant phages that evaded bNACHT32 (Escaper, ESC X.Y, where *X* indicates the parent phage number and *Y* indicates the escaper phage number). Escaper mutants were derived from one of three WT parent lineages and selected on bacteria expressing bNACHT32. RNA genome mutations that did not result coding changes are lower-case and the number represents the nucleotide within the gene/locus. Coding mutations are indicated as capital letters and the number represents the amino acid position within the protein.

†Indicates mutation is located in overlapping region between lys and rep genes.

abrogated cell death and DNA degradation phenotypes, demonstrating that translation of CP is required to activate bNACHT25 (Fig 2G and 2H). Expression levels of CP were confirmed for each of these mutants by western blot (S3A Fig). Despite the similar structure to CP from MS2 [27], CP$^{Qβ}$ only modestly induced DNA degradation and did not impact colony formation (Fig 2G and 2H). These findings suggest a mechanism for why bNACHT25 provides resistance against MS2 but not Qβ or Qβ-like phages.

We next tested if bNACHT25 could be activated by capsid proteins from dsDNA phages that bNACHT25 provides resistance to. The major and minor capsid proteins of phage T4, gp23 and gp24, and the major capsid protein of phage λ, gpE, did not inhibit colony formation in the presence of bNACHT25 (Fig 2G) [28,29]. These findings are, perhaps, unsurprising as CP from MS2 adopts a fold unique to the ssRNA phages that is not related to the HK97 fold capsid proteins from tailed dsDNA phages [30–32].

The other three proteins encoded by MS2 were also interrogated for activation of bNACHT25. Interestingly, significant DNA degradation was observed upon induction of the lysis protein L (S4 Fig). A slight amount of activation was also observed upon induction of genes encoding MP and Rep that resembled expression of CP$^{Qβ}$ (Figs 2H and S4). The relevance of these data is unclear as expression of L is extremely toxic to the bacteria. Of note, the inducible vector in the genetic assay expressed CP to higher than levels observed during MS2 infection (S3B Fig). Our findings suggest that synthesis of CP is required for activation of bNACHT25, and assembly of higher-order CP oligomers may be playing a role in the activation. Additionally, our data suggest that while bNACHT25 has specificity for recognizing synthesis of certain phage proteins, such as CP and L, it is possible that multiple proteins can activate the defense system during phage infection.

## CP does not directly interact with bNACHT25

We next sought to understand the mechanism of CP activation of bNACHT25. AVAST systems encode central STAND NTPase modules that are related to NACHT modules and are activated by binding of phage proteins [13,14,33]. We tested whether CP similarly activates bNACHT25 by direct binding using immunoprecipitation (IP) assays. CP failed to IP from bacteria expressing epitope-tagged, catalytically inactive bNACHT25 (FLAG-bNACHT25$^{D48A}$) (Fig 3A). bNACHT25$^{D48A}$ lacking a FLAG-tag was used as a negative control and the nuclease domain was catalytically inactivated to enrich for active complexes in the bacterial cytoplasm and prevent death of bacteria.

Our genetic data unambiguously showed bNACHT25 activation upon CP synthesis (Fig 2G and 2H), yet the two proteins did not form a complex (Fig 3A). We therefore hypothesized that bNACHT25 may instead be activated by a host protein that specifically changes during phage infection. To address this hypothesis, we performed mass spectrometry (MS) on tryptic digests of bNACHT25 IP samples to identify interacting proteins. A large cohort of proteins increased in

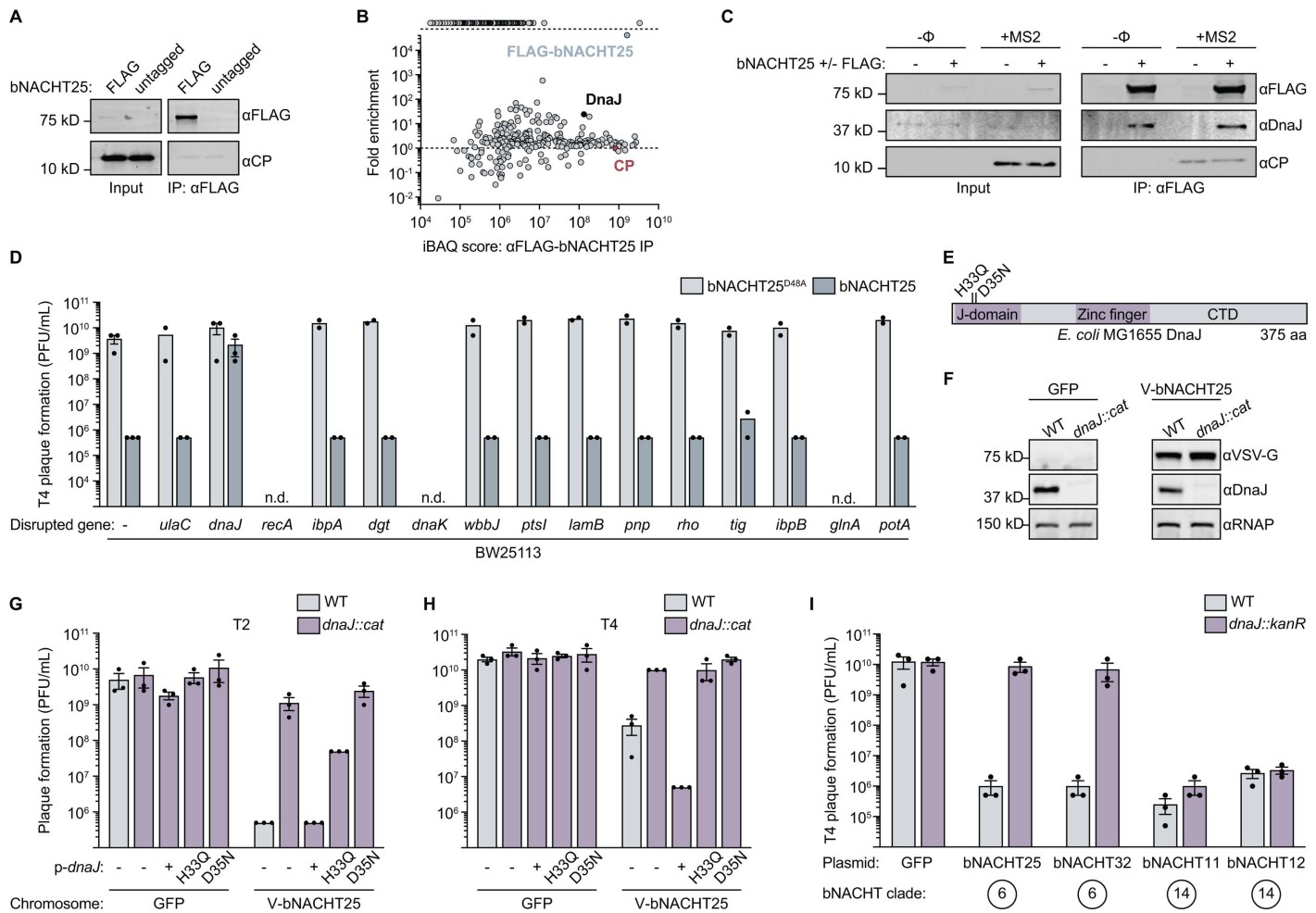

**Fig 3. Phage protection by bNACHT25 requires host *dnaJ*. (A)** Western blot analysis of αFLAG immunoprecipitation (IP) from *E. coli* expressing bNACHT25$^{D48A}$ either with an N-terminal FLAG tag or an untagged allele and CP from an inducible promoter. Input samples represent bacterial lysates prior to IP. Data are representative images of $n$ = 3 biological replicates. **(B)** Mass spectrometry (MS) of IP'd FLAG-bNACHT25. Data are intensity-based absolute quantification (iBAQ) score and fold enrichment comparing αFLAG IP from bacteria expressing bNACHT25$^{D48A}$ either with an N-terminal FLAG tag or an untagged allele. Proteins only detected in FLAG-bNACHT25 IP are represented as data points positioned above the top of the $y$-axis (dashed line). Data is a representative plot of $n$ = 2 biological replicates. See S2 Table for full MS results. **(C)** Western blot analysis of αFLAG IP from *E. coli* expressing bNACHT25$^{D48A}$ either with an N-terminal FLAG tag or an untagged allele either infected with MS2 at an MOI of 2 or without phage infection (−Φ). Input samples represent bacterial lysates prior to IP. Data are representative images of $n$ = 3 biological replicates. **(D)** Efficiency of plating of phage T4 on BW25113 mutants containing plasmids expressing bNACHT25 or bNACHT25$^{D48A}$. Disrupted genes are in order of abundance with the gene encoding the proteins with the highest iBAQ score listed first. Not determined (n.d.) indicates the strain was unable to grow with one or both plasmids. Data plotted as in Fig 1B. **(E)** Domain architecture of *E. coli* DnaJ labelled with mutants analyzed in Figs 3G–3H, S5A and S5B. **(F)** Western blot analysis of cell lysates generated from *E. coli* with the indicated genotypes. VSV-G-bNACHT25 (V-bNACHT25). Data are representative images of $n$ = 3 biological replicates. **(G–H)** Efficiency of plating of phage T2 **(G)** or T4 **(H)** on wild-type (WT) or *dnaJ::cat E. coli* MG1655 expressing either GFP or VSV-G-bNACHT25 from the chromosome. Complementation was achieved with p-*dnaJ*, which expresses *dnaJ* from an inducible promoter. Data plotted as in Fig 1B. *cat*: chloramphenicol acetyltransferase. **(I)** Efficiency of plating of phage T4 on BW25113 WT or *dnaJ::kanR* expressing the indicated defense system. Data plotted as in Fig 1B. The data underlying this figure can be found in S1 Data.

abundance over the negative control (Fig 3B and S2 Table). This analysis also confirmed bNACHT25 was successfully immunoprecipitated, while CP was not enriched (Fig 3B). We confirmed that during MS2 infection, CP did not IP with bNACHT25, while one of the top IP-MS hits, DnaJ, specifically pulled down with tagged bNACHT25 (Fig 3C).

## bNACHT25 activation is mediated by the host chaperone DnaJ

To identify the specific host protein required for bNACHT25-mediated sensing of phage, we introduced plasmids expressing bNACHT25 or the bNACHT25$^{D48A}$ catalytically inactive control into *E. coli* BW25113 carrying marked deletions of genes encoding each of 15 most enriched, non-essential IP-MS hits that repeated in both of our biological replicates [34]. These strains were then challenged with phage T4. Only a mutation in *dnaJ* impacted bNACHT25-dependent phage resistance (Fig 3D). These results suggest that DnaJ is required for bNACHT25 to protect against phage.

DnaJ is an Hsp40-family chaperone whose J domain is conserved from humans to bacteria [35]. DnaJ combats misfolded proteins in the cell in two ways. First, DnaJ can bind misfolded proteins and prevent their aggregation [36]. Second, DnaJ can deliver misfolded proteins to DnaK, an Hsp70-family chaperone, and stimulate the ATPase activity of DnaK through the J domain of DnaJ [37]. DnaJ has many substrates and is famously required for replication of phage λ and certain plasmids [38,39].

To understand the role of DnaJ in bNACHT25-mediated phage resistance, we constructed a marked *dnaJ* deletion (*dnaJ::cat*) in bacteria expressing VSV-G-bNACHT25 (VSV-G epitope tag introduced in order to visualize bNACHT25 expression) or a GFP negative control from the chromosome. Western blots confirmed *dnaJ* deletion and that the level of bNACHT25 was unperturbed by loss of *dnaJ* (Fig 3F). bNACHT25 no longer protected against the phages T2 and T4 in the absence of *dnaJ* (Fig 3G and 3H). Phage protection provided by bNACHT25 was rescued upon complementation with an inducible vector expressing *dnaJ* (Fig 3G and 3H). Interestingly, mutations in the J-domain of *dnaJ* that have previously been shown to disrupt binding between DnaJ and DnaK (D35N) [40] or the ability of DnaJ to stimulate the ATPase activity of DnaK (H33Q) [41] were not sufficient to rescue bNACHT25 phage defense in *dnaJ::cat* cells (Fig 3E, 3G and 3H). These mutants were expressed to the same level as wild-type DnaJ (S5A Fig). This finding suggests that the cochaperone activity of DnaJ in the DnaJ–DnaK chaperone cycle is required for phage protection by bNACHT25.

Similar results were obtained with VSV-G-bNACHT32 expressed from the chromosome, however bNACHT32 expression was slightly decreased in the *dnaJ::cat* background (S5B and S5C Fig). Together, these data suggest that bNACHT25, and potentially bNACHT32, require host DnaJ to protect against phage.

We next probed the specificity of phage detection through DnaJ by introducing plasmids expressing bNACHT proteins from clade 6 (bNACHT25 and bNACHT32) and clade 14 (bNACHT11 and bNACHT12) into wild-type and *dnaJ::kanR* bacteria. While bNACHT25 and bNACHT32 required *dnaJ* to protect against T4, bNACHT11 and bNACHT12 were unaffected by the disruption of *dnaJ* (Fig 3I), demonstrating that clade 14 bNACHT proteins have a different activation mechanism. We also investigated the specificity of host protein impacts on bNACHT25 by knocking out other *E. coli* chaperones, chaperone-related proteases, cold shock proteins, and phage shock proteins. bNACHT25 was expressed in bacteria deleted for *clpA/X*, *lon*, *hslO*, *spy*, *htpG*, *cspA/B/C*, *hscA/B*, and *pspB/C* and these strains were challenged with T4. bNACHT25 provided the same level of phage resistance for each of these as was observed in a wild-type background (S6 Fig). These findings confirm the specific role for DnaJ in bNACHT25-mediated phage defense.

Our data showed that phage protection mediated by bNACHT25 requires DnaJ and that bNACHT25 abundance is not impacted by loss of *dnaJ* (Fig 3F). However, since DnaJ is a chaperone that helps maintain protein solubility and function through interacting with exposed hydrophobic regions, an alternative hypothesis to explain our data is that DnaJ-deficient cells cannot express or fold fully functioning bNACHT25, resulting in *dnaJ*-dependence. Knocking out *dnaJ* did not cause a decrease in bNACHT25 abundance in whole cell lysates (S7A Fig). We next tested whether bNACHT25 becomes insoluble in the absence of *dnaJ* by separating whole cell lysates expressing bNACHT25 into soluble and insoluble fractions using centrifugation and testing for bNACHT25 abundance in the soluble fraction. We found that deleting *dnaJ* had no effect on the abundance of soluble bNACHT25 (S7A Fig). This result suggests that knocking out *dnaJ* does not decrease the levels of soluble bNACHT25.

   

To further show that bNACHT25 can function in the absence of DnaJ, we used a previously-characterized allele of bNACHT25 with a H506L mutation in the NACHT domain predicted to destabilize the ADP-bound, "OFF"-state of the protein, leading to stimulus-independent hyperactivation [9,43]. Induced expression of this hyperactive bNACHT25 allele led to DNA degradation in wild-type and *dnaJ*-mutant *E. coli*, with only a very small decrease in activity in early time-points (Fig 4A and 4B). The hyperactive mutant was expressed to the same level as wild-type (S7B Fig). These data demonstrate that downstream effector activation in a stimulus-independent system can still occur in the absence of *dnaJ*, confirming that bNACHT25 does not require DnaJ to adopt a functional conformation. These data help to rule out the hypothesis that DnaJ is simply required for bNACHT25 function and instead support a model where DnaJ is required for detection of phage protein synthesis upstream of bNACHT25 activation.

## DnaJ is required for activation of bNACHT25 by MS2 proteins

We were not able to measure changes to MS2 phage protection because MS2 failed to form plaques in bacteria lacking *dnaJ* (S8 Fig). However, we hypothesized that if DnaJ was required for MS2 detection, it would also be required for bNACHT25 activation by CP in our genetic assay. Accordingly, coexpression of CP with bNACHT25 resulted in DNA degradation in wild-type bacteria, but this phenotype was completely abrogated in a *dnaJ::cat* background (Fig 4C). Bacterial cell death on plates (Fig 4D) and in liquid culture (Fig 4E) caused by bNACHT25 and CP coexpression was also

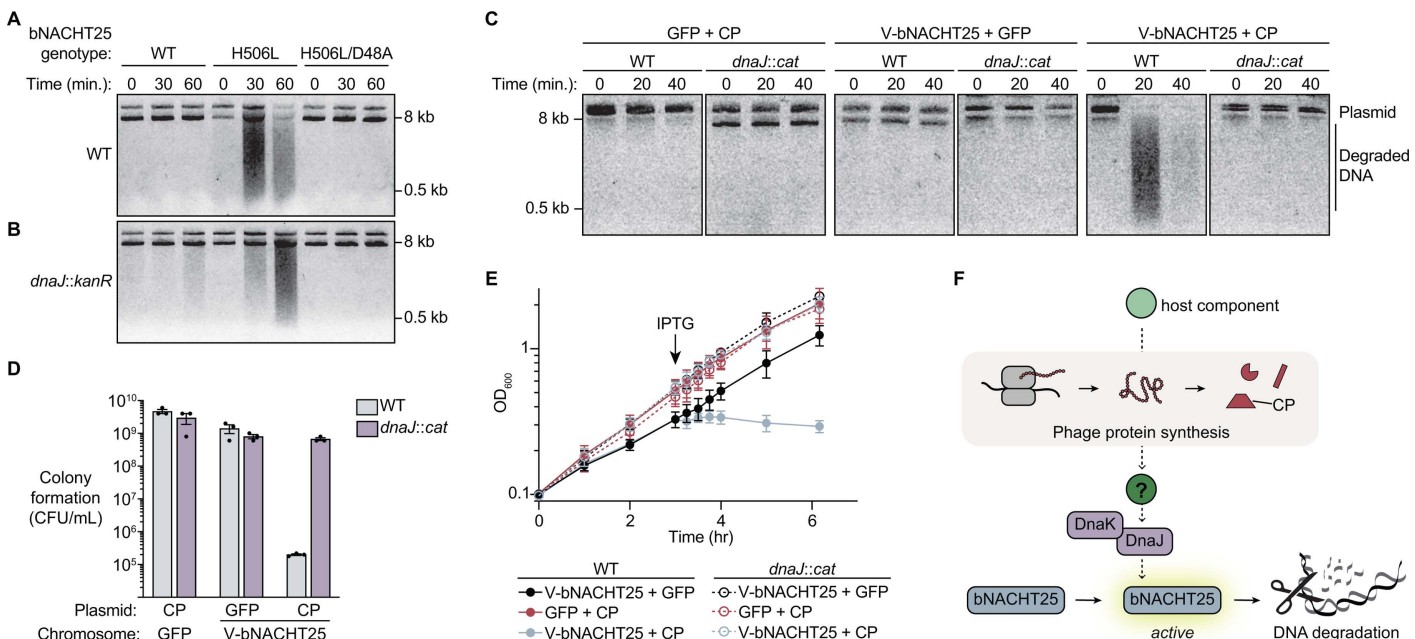

**Fig 4. bNACHT25 activation by phage proteins is mediated by DnaJ.** (A–B) Visualization of plasmid integrity in WT **(A)** or *dnaJ::kanR* **(B)** *E. coli* at the indicated timepoints post-induction of bNACHT25 expression with arabinose. Data are representative images of *n* = 3 biological replicates. **(C)** Visualization of plasmid integrity in *E. coli* with the indicated genotype at the indicated timepoints post-induction of CP with IPTG. bNACHT25 is on the chromosome and CP is on a plasmid. Data are representative images of *n* = 3 biological replicates. **(D)** Colony formation of *E. coli* with the indicated genotype following IPTG induction of GFP or CP. Data plotted as in Fig 2G. **(E)** Growth curves of *E. coli* expressing either GFP or V-bNACHT25 on the chromosome and either GFP or CP from an IPTG-inducible plasmid. Arrow indicates time point at which cultures were induced with IPTG. Data are the mean ± standard error of the mean (SEM) of *n* = 3 biological replicates. **(F)** Proposed model. Our data suggests that bNACHT25 monitors changes in the host that occur upon synthesis of phage proteins, such as CP, in a process that requires the chaperone DnaJ. The data underlying this figure can be found in S1 Data.

dependent on the presence of *dnaJ*. Expression levels of CP and bNACHT25 were comparable in wild-type and *dnaJ::cat* strains (S9 Fig). The inability of phage proteins to activate bNACHT25 in the absence of DnaJ is dramatic compared to the slight decrease in activity of the hyperactive allele of bNACHT25 without DnaJ.

Our analysis of MS2 proteins that activated bNACHT25 unexpectedly revealed that L, in addition to CP, activated nuclease activity (S4 Fig). L is the single-gene lysis protein of MS2 and was previously shown to bind to DnaJ, and *dnaJ* mutations limited host lysis by MS2 [42]. We hypothesized that L interactions with DnaJ were responsible for L-mediated bNACHT25 activation. Expression of L in a wild-type background induced strong activation of bNACHT25, and this activation was completely ablated in a *dnaJ::cat* background (S10 Fig), indicating that DnaJ is necessary for activation of bNACHT25 by both CP and L.

Taken together, these data support a model in which DnaJ is required for transmitting an activation signal from phage proteins to bNACHT25, suggesting that bNACHT25 guards host processes or proteins that specifically involve DnaJ (Fig 4F).

## Discussion

In this study we investigated the mechanism by which bacterial NLR-related proteins are activated by DNA and RNA phages. bNACHT25 protects against multiple ssRNA phages belonging to the *Emesvirus* genus and the model ssRNA phage MS2 was used to identify phage proteins that activate bNACHT25. We found that CP, the capsid protein of MS2, was sufficient to activate bNACHT25, however, this activation was not mediated by CP binding bNACHT25. Instead, our data suggests that CP synthesis induced changes to the cell that indirectly activated bNACHT25 in a process that depended on the host cell chaperone DnaJ. In addition, *dnaJ* was required for defense against the dsDNA phages T2 and T4, and mutations that disrupt the interaction between DnaJ and its cochaperone, DnaK, also disrupted bNACHT25-mediated defense. CP may be one of several diverse phage proteins produced at high levels during infection that stimulate the DnaJ/K-dependent bNACHT25 activation pathway. Our data illustrate an effective strategy used by bNACHT25 to detect a wide range of phages that do not share homology between any of their proteins.

A limitation of this study is that we have not uncovered the complete molecular mechanism for how CP expression leads to DnaJ-dependent activation of bNACHT25. CP did not form a complex with DnaJ and bNACHT25. Therefore, we are unable to say what the substrate is for DnaJ/K. We have ruled out the alternative hypothesis that *dnaJ* is required for expression of soluble and functional bNACHT25 by measuring expression levels of soluble wild-type bNACHT25 and showing that stimulus-independent bNACHT25 is largely able to degrade DNA. We believe that DnaJ forms a complex with bNACHT25 and potentially other proteins to sense phage infection. However, we cannot rule out that DnaJ is required for proper folding or function of another, upstream protein involved in bNACHT25 phage sensing. The specific interactions that lead to activation of bNACHT25 by CP remain an exciting area of future investigation.

Two dominant mechanisms innate immune pathways use to sense invading pathogens are to detect either (1) pathogen-derived molecules or (2) pathogen-specific activities [2,44,45]. In the first mechanism, pathogen-associated molecular patterns (PAMPs) are directly sensed by pattern recognition receptors (PRRs). The PAMP–PRR model relies on PAMPs being invariant and crucial to the pathogen to limit immune evasion [46]. In the second mechanism, often termed effector-triggered immunity or the guard model, the innate immune system monitors for the biochemical activity of a pathogen-encoded effector [47]. The guard model relies on the function of pathogen effectors being essential to causing disease to limit immune evasion. NLRs in eukaryotes exemplify both of these mechanisms. The human NAIP/NLRC4 inflammasome is a PRR that detects structural components of bacterial type 3 secretion systems as a PAMP [48]. The *Arabidopsis* RPM1 R-protein is a guard for RIN4 and is activated by the pathogen effectors AvrRpm1 and AvrB [49]. We propose that bNACHT25 adopts a mechanism of sensing most similar to the guard model and senses perturbations to the host cell proteome. CP is not a conventional pathogen effector; however, it still impacts a core host process that bNACHT25 monitors as a proxy for infection. Across bacteria, bNACHT25 joins retrons [50,51] and ToxIN [52] in a growing list antiphage systems that indirectly survey the cell for signs of infection [1,2].

We propose that bNACHT25 is activated by binding DnaJ in complex with substrate proteins that become available during phage infection. Those substrates may be from the host cell as a result of CP expression or L protein itself interacting with DnaJ. In this way, bNACHT25 can guard both DnaJ and upstream host chaperones (e.g., DnaK or GroEL/ES) whose occupancy by CP may result in increased unfolded proteins. It is not clear what CP expression may specifically be doing to the cell, however, there are many lines of evidence linking viral infection and host chaperones. Several phages, including λ and T4, require the GroEL chaperonin to fold their capsid proteins [53,54]. Additionally, λ requires a functional DnaJ/DnaK chaperone system to complete its replication cycle [38], and MS2 experiences delayed lysis in a DnaJ mutant background, likely due to interactions between L and DnaJ [42]. The dependence of some phage proteins, such as capsids, on chaperones to properly fold and assemble is curious. It suggests that perhaps there is a trade-off that a virus must make: in order to adopt the ideal capsid structure, they must use a host chaperone to assist in protein folding. Consistent with this, conservative point mutations in the major capsid protein of phage T4 can bypass the GroEL chaperonin, but these are not equivalently well-expressed to wild-type in GroEL proficient strains [55]. Intriguingly, T4 bypass mutants that no longer require GroEL are reminiscent of the CP escaper mutations identified here.

The requirement of phages for host chaperones provides the cell with a unique opportunity to detect a wide range of viruses through DnaJ without an easy path for the phage to escape immune detection. Guarding host chaperones may therefore be an effective strategy used by other uncharacterized antiphage systems. Given the highly conserved nature of DnaJ/Hsp40-family proteins throughout life, it may further be true that eukaryotic innate immune pathways also monitor their DnaJ homologs.

This study is not the first time chaperones have been shown to play a role in the function of NLRs. The Hsp90 chaperone helps maintain the stability of the inactive form of NLRP3 in human cells, thus regulating its activity [56–58]. Hsp90 is also required for immunity conferred by several plant R proteins, often by helping to maintain appropriate levels of these sensors in the cell [59,60]. While the stability of bNACHT25 is not altered by the absence of DnaJ, DnaJ plays a crucial role its ability to recognize pathogen-derived signals. Hence, regulation of NLR activation by chaperones appears to be a conserved strategy across bacteria and eukaryotes.

## Materials and methods

### Bacterial strains and culture conditions

*E. coli* strains used in this study are listed in S3A Table. Unless otherwise indicated, all cultures were grown in 1–4 mL of media in 14 mL culture tubes shaking at 220 rpm at 37 °C. "Overnight" cultures were started from either a single colony or glycerol stock and grown for 16–20 h following inoculation. Culture media was supplemented with carbenicillin (100 µg/mL), chloramphenicol (20 µg/mL), kanamycin (50 µg/mL), and/or tetracycline (15 µg/mL) when applicable for plasmid maintenance or strain verification. Experiments were performed with *E. coli* MG1655 (CGSC6300) or *E. coli* BW25113 (CGSC7636). *E. coli* OmniPir [9] was used for construction and storage of plasmids. Where indicated, bNACHT sequences were inserted into the *lacZ* locus of *E. coli* MG1655 using Lambda red methodology as previously described [9,61].

Bacterial cultures used for cloning, strain construction, indicated colony formation assays, and immunoprecipitation assays were grown in LB medium (1% tryptone, 0.5% yeast extract, and 0.5% NaCl). Strains were frozen for long-term storage in LB supplemented with 30% glycerol at −70 °C. Bacteria used to perform phage propagation, phage infection assays, DNA degradation assays, and indicated colony formation assays were cultivated in "MMCG" minimal medium (47.8 mM $Na_2HPO_4$, 22 mM $KH_2PO_4$, 18.7 mM $NH_4Cl$, 8.6 mM NaCl, 22.2 mM glucose, 2 mM $MgSO_4$, 100 mM $CaCl_2$, 3 mM thiamine, Trace Metals at 0.01% v/v (Trace Metals mixture T1001, Teknova, final concentration: 8.3 µM $FeCl_3$, 2.7 µM $CaCl_2$, 1.4 µM $MnCl_2$, 1.8 µM $ZnSO_4$, 370 nM $CoCl_2$, 250 nM $CuCl_2$, 350 nM $NiCl_2$, 240 nM $Na_2MoO_4$, 200 nM $Na_2SeO_4$, 200 nM $H_3BO_3$)). When a strain with two plasmids was cultivated in MMCG medium, bacteria were grown in carbenicillin (50 µg/mL) and chloramphenicol (10 µg/mL). MMCG and LB agar plates contain 1.6% agar and media components described above.

## Conjugation of F′ plasmid into *E. coli* MG1655

The plasmid F′ was introduced into various strains of *E. coli* MG1655 via conjugation. Briefly, 400 µL of overnight cultures of donor (*E. coli* MG1655 + F′ donor grown in LB + tetracycline) and recipient (various *E. coli* strains grown with proper antibiotics) were pelleted at 10,000 × *g* for 1 min. Pellets were washed in 20 µL of LB, pelleted again, and resuspended in 20 µL of LB. Resuspensions of donor alone, recipient alone, and mixture of donor and recipient were spotted on an LB plate, which was incubated upright for 1 h at 37 °C. Spots were then struck to single colonies on LB + tetracycline + appropriate antibiotics to select for recipients that had received F′.

## Plasmid construction

Plasmids used in this study are listed in S3A Table. MS2 and Qβ phage genes were amplified from synthesized cDNA (see below). T4 and λ genes were amplified from 1 µL of boiled, diluted phage lysate. T4 gp23* and gp24* include amino acids 66-520 and 11-426, respectively, which are the regions of gp23 and gp24 that are maintained post-proteolytic cleavage [30]. bNACHT genes were cloned from plasmids previously generated [9]. *E. coli* genes were cloned from purified *E. coli* MG1655 genomic DNA. See S3C Table for protein accession numbers.

bNACHT genes along upstream native promoter regions/downstream terminator regions were cloned into the SbfI/NotI sites of the pLOCO2 vector or the EcoRI/HindIII sites of the pBAD30x vector, as previously described [9]. All phage genes were cloned into the NotI and BamHI or BmtI sites of pTACxc (for chloramphenicol selection) or pTACx (for carbenicillin selection) vector for inducible expression with IPTG. Restriction enzymes were purchased from New England Biolabs. DNA sequences were amplified and amended with ≥18 bp homology to their destination vectors using Q5 Hot Start High Fidelity Master Mix (NEB, M0494L). Vectors were constructed using Gibson Assembly with HiFi DNA Assembly Master Mix (NEB, E2621L) as described [62] and transformed into OmniPir via heat shock or electroporation. VSV-G and 3X-FLAG tags were appended to the N-termini of bNACHT25 and bNACHT32 in order to perform western blots and IPs on these proteins. To add epitope tags, tag sequences were included in 3′ sequences of primers that annealed to the gene of interest. Plasmid sequences were verified with Sanger sequencing (Quintara Biosciences or Azenta).

## Phage amplification and storage

Phages used in this study are listed in S3B Table. F-dependent RNA phages were amplified using the *E. coli* MG1655 + F′ host [63,64], and dsDNA phages were amplified using *E. coli* MG1655. A modified double agar overlay plate amplification was used to generate phage lysates [65]. Briefly, 400 µL of mid-logarithmic phase ($OD_{600}$ = 0.2–0.8) bacterial cultures were combined with 5,000–15,000 plaque forming units (PFU) of phage and incubated for 1 min at room temperature to allow for adsorption. An amount of 3.5 mL of "top agar" (0.35% agar, 10 mM $MgCl_2$, 10 mM $CaCl_2$, and 100 µM $MnCl_2$, 0.01% v/v Trace Metals) was then added to the phage-bacteria mixture which was then plated directly onto an MMCG agar plate. Plates were incubated overnight at 37 °C. To recover the amplified phage from the top agar, 5 mL of SM buffer (100 mM NaCl, 8 mM $MgSO_4$, 50 mM Tris-HCl pH 7.5, 0.01% gelatin) was applied to the top of the plate and the plate was incubated for 1–6 h at room temperature. Phages were then harvested by transferring SM buffer from the plate to a tube. To increase phage titers, top agar overlay was sometimes scraped and harvested along with the SM buffer. In this case, the SM buffer was first centrifuged at 4,000 × *g* for 10 min and supernatant was transferred to a new tube. Two to three drops of chloroform were added to all phage stocks followed by vortexing to sterilize the lysate. Phages were stored at 4 °C.

MS2-like (FrSaffron, FrHenna, and FrBlood) and Qβ-like (FrSangria, FrMerlot, FrHibiscus, FrBurgundy) phages were kindly provided by Michael Baym and colleagues. To plaque purify, serial dilutions of these phage stocks were plated as described in the above paragraph, and single plaques were picked using a glass Pasteur pipet and soaked out in 500 µL SM buffer with 1–2 drops of chloroform.

We observed that MS2 and Qβ phage stocks were prone to loss of infectiousness over time at 4 °C, at times inexplicably losing over $10^4$ PFU within 6 months. To maintain phage stocks, MS2 and Qβ were frozen for long term storage according to published methods [66]. Briefly, phage was added to mid-logarithmic phase *E. coli* MG1655 + F′ in MMCG medium at an MOI of 0.5. The phage and bacteria were incubated for 15 min at room temperature without shaking to allow for adsorption. The mixture was then combined with glycerol (final concentration 15%), flash frozen in liquid nitrogen, and stored at −70 °C. To recover and amplify phages from frozen glycerol stocks, a small amount of stock was scraped and swirled into 100 μL of SM buffer. This mixture was then used in a plate amplification with *E. coli* MG1655 + F′ as described above.

## Phage infection assays estimated efficiency of plaque formation

To quantify efficiency of plaque formation or titer for phage lysates, including determining phage defense, 400 μL of mid-logarithmic phase MG1655 + F′ expressing the defense system from the indicated vector or from the chromosome was combined with 3.5 mL of top agar and plated onto an MMCG agar plate. Once solidified, 2 μL of 10-fold serial dilutions of phage in SM buffer were spotted onto the top agar overlay. Once dried, plates were incubated overnight at 37 °C. The next day, plaques were quantified. When single plaques were not distinguishable/countable, the most dilute spot with visible phage (i.e., a hazy zone of clearance) was recorded as 10 PFU. When no plaques or zone of clearance were visible at the most concentrated spot, plaque formation fell below the limit of detection and 0.9 PFU were recorded for that spot. Phage protection data was reported as PFU/mL ± standard error of the mean (S.E.M) of $n$ = 3 biological replicates.

## Phage infection time course in liquid culture

Bacteria containing the indicated plasmid were grown to mid-logarithmic phase in 25-mL MMCG cultures. An amount of 150 μL of cultures $OD_{600}$-normalized to 0.15 were added to wells of a 96-well plate in triplicates. Phage was added to each bacterial culture at the indicated MOI. Final volume of phage added to each well was 5 μL. Beginning immediately after infection, $OD_{600}$ readings were recorded every 2 min over the course of the experiment using the SPARK Plate Reader (TECAN). Bacterial growth curves are representative of $n$ = 3 biological replicates. The mean of 3 technical replicates was plotted for each time point.

## Knockout strains used for testing bNACHT25-mediated phage protection

Gene deletion mutants described in Figs 3D and S6 were constructed by replacing the indicated gene with the kanamycin resistance cassette (*kanR*), as previously described (Keio collection) [34] and obtained as glycerol stock duplicates. Strains were streaked to single colonies on LB + kanamycin, cultivated overnight, and transformed with plasmids expressing bNACHT25 or bNACHT25[D48A] via electroporation.

## MS2 escaper mutant generation

Escaper mutants were generated from 3 independent, wild-type parent MS2 phage stocks that were separately plate amplified on *E. coli* lacking a defense system. To generate escaper phages, approximately $5 \times 10^6$ PFU of each parent phage was used to infect *E. coli* carrying plasmids expressing either bNACHT02, bNACHT12, bNACHT25, or bNACHT32 using the modified double agar overlay assay described above. Six single plaques were picked from each amplification ("primary escapers"), soaked out in 500 μL of SM buffer, and re-amplified on bacterial lawns expressing the corresponding defense system to repurify and confirm escaper mutants ("secondary escapers"). bNACHT02 and bNACHT12 primary escapers were diluted 1:1000 to generate single secondary escaper plaques. bNACHT32 primary escapers required higher concentrations of phage lysates to obtain secondary escapers plaques (undiluted or 1:100 dilutions), and no bNACHT25 primary escapers generated plaques upon repurification. Single secondary escaper plaques were picked and

soaked out in SM buffer for storage. The resulting 28 escaper phage lysates were plate amplified and escapers of interest were spot plated onto bacteria expressing the corresponding bNACHT system to confirm escaper phenotype.

### Phage genome sequencing and escaper analysis

RNA genomes of MS2 and Qβ were purified as previously described [9]. Briefly, phage lysates were treated with DNase I, and RNA was extracted using the PureLink RNA Minikit (Invitrogen) according to the manufacturer's instructions. On-column DNase treatment was omitted, and RNA was eluted in 30 μL of nuclease-free water. RNA phage cDNA was synthesized using the Invitrogen SuperScript III First-Strand Synthesis System as previously described [9]. MS2 cDNA was then PCR amplified in 3 overlapping fragments using OneTaq polymerase using previously described primers [67]. Amplified MS2 genome was prepared for Illumina sequencing using a modification of the Nextera kit protocol as previously described [68]. Samples were sequenced using the Illumina MiSeq V2 Micro 300-cycle kit (CU Anschutz Genomics and Microarray Core). Using the Map to Reference feature in Geneious software, reads were mapped to the MS2 reference genome (NCBI Genome accession NC_001417). Under the Trim Primers option, the Nextera Trimming Oligo (AGATGTG-TATAAGAGACAG) was trimmed from reads; otherwise, default settings were used.

The Geneious feature "Find Variations/SNPs" was used to identify variants in escaper genomes. Variants were identified as escaper mutations if they were present in ≥70% of reads and were not present in parent genomes.

### bNACHT25 (DNA degradation) activation assay

Bacteria containing the indicated plasmids were grown to mid-logarithmic phase in 30 mL MMCG containing appropriate antibiotics. If applicable, phage was added at an MOI of 2, or 500 μM IPTG or 0.2% (v/v) arabinose was added to induce gene expression. Following infection/induction, $2 \times 10^9$ CFU were harvested from each strain by pelleting at $4,000 \times g$ for 10 min at 4 °C. Plasmid DNA was extracted using standard miniprep protocol (Qiagen) and eluted in 60 μL DNA elution buffer (10 mM Tris-HCl pH 8.0). Twenty microliters of eluted DNA was combined with 5 μL of 6× Loading Dye (final concentration 4 mM Tris-HCl, 3% Ficoll-400, 12 mM EDTA, 0.04% Orange G, pH 8) and run for 30 min at 130 V on a 1% agarose gel (1% agarose, 40 mM Tris, 20 mM acetic acid, 1 mM EDTA, 0.02% v/v SYBR Safe DNA stain). Gels were imaged using an Azure Biosystems Azure 200 Bioanalytical Imaging System.

### Colony formation assays for measuring growth inhibition

To test for bacterial growth inhibition, 5 μL of 10-fold serial dilutions of indicated cultures were spotted onto the appropriate agar plates. For testing bNACHT32 activation in Fig 2D, overnight cultures in MMCG were spotted onto MMCG plates carbenicillin (20 μg/mL) and chloramphenicol (4 μg/mL) with or without 500 μM IPTG. In all other cases, overnight cultures in LB were spotted onto LB plates containing appropriate antibiotics and either 0.2% glucose for uninduced conditions or 500 μM IPTG for induced conditions. Once dry, plates were incubated overnight at 37 °C. The next day, colonies were quantified. In instances where single colonies were not distinguishable at the most dilute spot with visible bacteria (i.e., hazy spot of bacteria), 10 colony forming units (CFU) were recorded. When no colonies were visible at the most concentrated spot, 0.9 CFU were recorded. Colony formation data was reported as CFU/mL ± S.E.M of $n = 3$ biological replicates.

### Liquid culture time course for measuring growth inhibition

Bacteria containing the indicated plasmids were diluted to an $OD_{600}$ of 0.1 in 30 mL MMCG containing appropriate antibiotics and cultivated shaking at 37 °C. When cultures reached mid-logarithmic phase ($t = 3$ h), 500 μM IPTG was added to induce gene expression. $OD_{600}$ measurements were taken at the indicated timepoints over the course of the experiment. Growth curves are reported as the mean ± S.E.M of $n = 3$ biological replicates.

## Construction of *dnaJ* knockout strains

The *E. coli* MG1655 *dnaJ* gene was replaced using Lambda red recombination with *cat*, a gene encoding chloramphenicol acetyltransferase, as described previously [61]. Briefly, *cat* and its promoter were amplified from pKD3 with overhangs containing homology to 50 bp immediately upstream and downstream of the *dnaJ* gene. *E. coli* with chromosomally inserted KanR-VSVG-bNACHT25, KanR-VSVG-bNACHT32, or KanR-GFPmut3 carrying the pKD46 plasmid were grown to mid-logarithmic phase in LB + 0.2% arabinose to induce the Lambda red system. These bacteria were then made electrocompetent, transformed with gel-purified PCR products containing the *cat* insert via electroporation, and recovered for 2 h at 30 °C. Cultures were then plated on LB+ chloramphenicol to select for acquisition of *cat*. PCR and western blot were used to confirm the replacement of *dnaJ* with *cat* (Figs 3F and S5C).

## Western blots

To measure protein expression via western blot, bacterial strains expressing indicated gene(s) were grown to mid-logarithmic phase. When applicable, 500 μM IPTG or phage at an MOI of 2 was added, and cultures were grown for an additional 20 min or indicated timepoints before extracting whole cell lysates. To harvest, $5 \times 10^8$ CFU were pelleted and when testing for CP expression after infection with MS2, pellets were washed with 1 mL of sterile water and re-pelleted. Pellets were all resuspended in 50 μL of LDS buffer (106 mM Tris-HCl pH 7.4, 141 mM Tris Base, 2% w/v lithium dodecyl sulfate, 10% v/v glycerol, 0.51 mM EDTA, 0.05% Orange G). Samples were then incubated at 95 °C for 10 min and centrifuged at $20,000 \times g$ for 3–5 min to remove debris. Samples in LDS were loaded at equal volumes on a 4%–20% SDS–PAGE gel, run for approximately 1–2 h at 100–160 V, and transferred to PVDF membranes charged in methanol. Membranes were blocked in Licor Intercept Buffer for 1 h at room temperature and incubated with primary antibodies diluted in Intercept buffer overnight at 4 °C with rocking. αVSV-G (Rockland, RRID:AB_217930) was used at 1:10,000 to detect VSV-G-bNACHT25 and VSV-G-bNACHT32, αCP (Millipore, RRID:AB_2827507) was used at 1:10,000 to detect CP, αFLAG (Sigma, RRID:AB_259529) was used at 1:10,000 to detect 3× FLAG-bNACHT25$^{D48A}$, and αDnaJ (Enzo, RRID:AB_2039063) was used at 1:2,500 to detect DnaJ. αRNAP (BioLegend Cat# 663,006, RRID:AB_2565555) was used at 1:5,000 as a loading control for testing protein levels.

Membranes were washed 3 times for 10 min each in TBS-T (Tris-buffered saline, 0.1% Tween-20) then incubated in Licor infrared (800CW/680RD) αRabbit/Mouse secondary antibodies diluted 1:40,000 in TBS-T for 1 h at room temperature. Blots were visualized with the Licor Odyssey CLx.

## Immunoprecipitation (IP) assays

Immunoprecipitation assays were performed as described previously [69]. Briefly, *E. coli* MG1655 expressing 3× FLAG-bNACHT25$^{D48A}$ or bNACHT25$^{D48A}$ from one plasmid and inducible CP from another plasmid were grown in 25 mL of LB with 500 μM IPTG until they reached mid-logarithmic phase. OD$_{600}$-normalized cultures (approximately $1 \times 10^{10}$ CFU) were then centrifuged to $4,000 \times g$ for 10 min at 4 °C. The resulting pellet was resuspended in 4.5 mL lysis buffer (400 mM NaCl, 20 mM Tris-HCl pH 7.5, 2% glycerol, 1% triton, and 1 mM β-mercaptoethanol). Cells were lysed by sonication and then centrifuged at $20,000 \times g$ and 4 °C to remove cellular debris. Fifty microliters of each soluble lysate was saved as "Input." Samples were then mixed with 30 μL magnetic beads covalently linked to the αFLAG M2 antibody (Sigma) overnight at 4 °C with end-over-end rotation. After 3 washes in lysis buffer, IP beads were resuspended in 50 μL LDS. SDS–PAGE and western blots were performed as described above.

## Mass spectrometry (MS) analysis

An IP was performed as described above with the following alterations. To increase protein yields, 100 mL of induced, logarithmic-phase culture was processed for each sample. Following incubation of bacterial lysates with

αFLAG beads, beads were washed 3 times with 20 mM ammonium bicarbonate. Dry beads were then subjected to on-bead trypsin digest followed by analysis on a Thermo Obitrap Q-Exactive HF-X using nanoLC–MS MS, as previously described [69]. Peptides were mapped to the proteome of *E. coli* MG1655 (https://uniprot.org/proteomes/UP000000625) in addition to MS2 CP and 3× FLAG-bNACHT25$^{D48A}$ sequences. Two biological replicates of the IP were performed and analyzed by MS. Data from replicate 1 can be found in S2A Table and replicate 2 can be found in S2B Table.

## bNACHT25 solubility assay

*E. coli* MG1655 wild-type or *dnaJ::cat* expressing VSV-G-bNACHT25 from the bacterial chromosome were grown to mid-logarithmic phase in 25 mL of LB medium. $OD_{600}$-normalized cultures (approximately $1 \times 10^{10}$ CFU) were then centrifuged to $4,000 \times g$ for 10 min at 4 °C. The resulting pellet was resuspended in 4.5 mL lysis buffer (400 mM NaCl, 20 mM Tris-HCl pH 7.5, 2% glycerol, 1% triton, and 1 mM β-mercaptoethanol). Cells were lysed by sonication, and 1 mL of this material was saved as "Whole Cell Lysate" (WCL). The remaining lysate and then centrifuged at $21,000 \times g$ and 4 °C for 30 min to remove insoluble components of the lysate. The supernatant was saved as "Soluble" (S). Western blots were performed as described above.

## Accession numbers

See S3C Table for complete list of protein accession numbers. Briefly, bNACHT02: WP_021557529.1, bNACHT12: WP_021519735.1, bNACHT25: WP_001702659.1, bNACHT32: WP_057688292, bNACHT11: WP_114260439.1, CP (MS2): YP_009640125.1, CP (Qβ): BAP18764.1, maturation protein (MS2): YP_009640124.1, lysis protein (MS2): YP_009640126.1, replicase (MS2): YP_009640127.1, gp23 (T4): AAD42428.1, gp24 (T4): AAD42429.1, gpE (λ): AAA96540.1, DnaJ (*E. coli* MG1655): NP_414556.1.

## Supporting information

**S1 Fig. Nucleotide sequence identity of *Emesvirus* and *Qubevirus* ssRNA phages.** Pairwise nucleotide identity (%) of ssRNA phages investigated in this study.
(PDF)

**S2 Fig. bNACHT25 protects against infection by MS2 but not Qβ in liquid culture. (A–D)** Growth curves of *E. coli* expressing either GFP or bNACHT25. $OD_{600}$ measurements began immediately following infection with the indicated phages at the indicated MOI. Data are representative of $n = 3$ biological replicates. The mean of $n = 3$ technical replicates for a representative experiment is shown. The data underlying this figure can be found in S1 Data.
(PDF)

**S3 Fig. CP expression levels. (A)** Western blot analysis of bacterial lysates generated from *E. coli* expressing the indicated CP alleles and GFP from the chromosome. Bacteria were harvested 20-min post-induction with IPTG. **(B)** Western blot analysis of *E. coli* lysates generated from strains expressing bNACHT25 harvested at the indicated timepoints following infection with MS2 at MOI of 2 or induction of CP with IPTG. For **(A–B)**, data are representative images of $n = 3$ biological replicates.
(PDF)

**S4 Fig. Analysis of MS2 genes with bNACHT25.** Visualization of plasmid integrity in *E. coli* expressing either GFP or bNACHT25 on the chromosome coexpressed with the MS2 protein indicated via an inducible plasmid. Plasmid DNA was harvested at indicated timepoints post-induction with IPTG. Data are representative images of $n = 3$ biological replicates.
(PDF)

**S5 Fig. DnaJ is required for bNACHT32 phage protection. (A)** Western blot analysis of cell lysates generated from *E. coli* with the indicated genotypes. p-*dnaJ* expresses *dnaJ* to WT levels without adding inducer. Data are representative images of *n* = 2 biological replicates. **(B)** Efficiency of plating of the indicated phage on WT or *dnaJ::cat E. coli* MG1655 expressing either GFP or VSV-G-bNACHT32 (V-bNACHT32) from the chromosome. Data plotted as in Fig 1B. Chloramphenicol acetyltransferase (*cat*). **(C)** Western blot analysis of cell lysates generated from *E. coli* with the indicated genotypes. Data are representative images of *n* = 3 biological replicates. The data underlying this figure can be found in S1 Data.
(PDF)

**S6 Fig. Screening bNACHT25 phage protection in *E. coli* chaperone mutants.** Efficiency of plating of phage T4 on *E. coli* BW25113 mutants containing plasmids expressing bNACHT25 or bNACHT25$^{D48A}$. Not determined (n.d.) indicates the strain was unable to grow with one or both plasmids in the assayed conditions. Data plotted as in Fig 1B. The data underlying this figure can be found in S1 Data.
(PDF)

**S7 Fig. Soluble bNACHT25 is not decreased by the loss of *dnaJ*. (A)** Western blot analysis of Whole Cell (WCL) and Soluble (S) lysates obtained from *E. coli* WT or *dnaJ::cat* expressing VSV-G-bNACHT25 from the bacterial chromosome. **(B)** Western blot analysis of cell lysates generated from *E. coli* BW25113 expressing VSV-G-bNACHT25 with the indicated genotypes. All data are representative images of *n* = 3 biological replicates.
(PDF)

**S8 Fig. Infection by diverse phages requires *dnaJ*.** Efficiency of plating of phages T4, λvir, MS2, and Qβ on BW25113 WT or *dnaJ::kanR*. Data are representative images of *n* = 3 biological replicates.
(PDF)

**S9 Fig. CP and bNACHT25 expression in wild-type and *dnaJ::cat* cells.** Western blot analysis of *E. coli* lysates from the indicated genotypes, timepoints, and conditions. Data are representative images of *n* = 3 biological replicates.
(PDF)

**S10 Fig. Activation of bNACHT25 by MS2 L protein requires DnaJ.** Visualization of plasmid integrity in *E. coli* with the indicated genotype at the indicated timepoints post-induction of L with IPTG. bNACHT25 is on the chromosome and L is on a plasmid. Data are representative images of *n* = 3 biological replicates.
(PDF)

**S1 Table. All MS2 escaper mutations identified.**
(XLSX)

**S2 Table. Mass spectrometry results for immunoprecipitated bNACHT25.**
(XLSX)

**S3 Table. Strains, plasmids, phages, and accession numbers used in this study.**
(XLSX)

**S1 Raw Images. Original blot and gel images for data in the main and supplemental figures.**
(PDF)

**S1 Data. Numerical values for graphs in the main and supplemental figures.**
(XLSX)

## Acknowledgments

The authors would like to thank M. Baym, S.V. Owen, and colleagues for generously sharing F-dependent phages; C. Ebmeier and the Mass Spectrometry Facility at CU Boulder for assistance with sample preparation, experimental details, and data analysis (NIH S10-OD025267); A. Erbse and the CU Boulder Department of Biochemistry Shared Instruments Pool core facility (RRID:SCR_018986) and its staff; Drew O'Brien for assistance with MS2 escaper generation; and members of the Whiteley lab for advice and helpful discussions.

## Author contributions

**Conceptualization:** Amy N. Conte, Aaron T. Whiteley.

**Data curation:** Amy N. Conte.

**Funding acquisition:** Aaron T. Whiteley.

**Investigation:** Amy N. Conte, Madison E. Ruchel, Samantha M. Ridgeway, Aaron T. Whiteley.

**Methodology:** Amy N. Conte, Emily M. Kibby, Aaron T. Whiteley.

**Project administration:** Aaron T. Whiteley.

**Resources:** Amy N. Conte, Emily M. Kibby, Toni A. Nagy, Aaron T. Whiteley.

**Supervision:** Aaron T. Whiteley.

**Validation:** Amy N. Conte.

**Visualization:** Amy N. Conte.

**Writing – original draft:** Amy N. Conte, Aaron T. Whiteley.

**Writing – review & editing:** Amy N. Conte, Madison E. Ruchel, Samantha M. Ridgeway, Emily M. Kibby, Toni A. Nagy, Aaron T. Whiteley.

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
