## [Editor Report · Decision Letter 0]

18 Dec 2024

Dear Aaron,

I hope all is well. Thank you for submitting your revised manuscript entitled "Phage detection by a bacterial NLR-related protein is mediated by DnaJ" for consideration as a Research Article by PLOS Biology.

Your manuscript has now been evaluated by the PLOS Biology editorial staff as well as by the previous academic editor and I am writing to let you know that we would like to send the revision back to the original reviewers.

Once your full submission is complete, your paper will undergo a series of checks in preparation for peer review. After your manuscript has passed the checks it will be sent out for review. To provide the metadata for your submission, please Login to Editorial Manager (https://www.editorialmanager.com/pbiology) within two working days, i.e. by Dec 20 2024 11:59PM.

IMPORTANT: PLOS Biology will be "closed" from December 23rd to January 2nd, I will probably be only able to invite the reviewers in January, which will delay the process. However, this will be in my priority list on my return to work.

Best wishes,

Melissa

Melissa Vazquez Hernandez, Ph.D.

Associate Editor

PLOS Biology

---

## [Decision Letter · Decision Letter 1]

4 Feb 2025

Dear Aaron,

Thank you for your patience while we considered your revised manuscript "Phage detection by a bacterial NLR-related protein is mediated by DnaJ" for publication as a Research Article at PLOS Biology. Your revised study has been evaluated by the PLOS Biology editors, the Academic Editor and three of the original reviewers.

We appreciate the work you have already done in this revision, which has been acknowledged by the reviewers. Both we and the reviewers find your study highly interesting; however, several concerns remain that must be addressed. Reviewer 2 (R2) suggests revising the language regarding the role of DnaJ in phage sensing, as it remains unclear. Reviewer 3 (R3) notes that the role of DnaJ is not supported by in vitro evidence, there is no indication whether bNACHT25 is soluble or insoluble in the absence of DnaJ, and the mechanism of sensing has not been described. Reviewer 4 (R4) highlights the lack of a proposed mechanism by which bNACHT detects the phage or blocks its infection. Additionally, R4 points out insufficient evidence that CP is sufficient for activation, binds to DnaJ, or that DnaJ mediates bNACHT activity.

After discussion with the Academic Editor, and considering the reviewers' interest, as well as our interest in the topic, we propose changing the article type to a Discovery Report. This format is designed for studies of high relevance that may not present a complete story, such as lacking a full molecular mechanism, but are expected to publish an Update Article in the future. With this change, we will not require the complete mechanism but do request stronger evidence as suggested by R3 and recommend tempering certain claims based on R2 and R4's feedback. Specifically, we strongly encourage addressing point #1 from R2's comments and require experimentally addressing point #2. Successfully addressing point #1 would likely support retaining the "sensing" claim. The Academic Editor also provided some feedback which you can find after my signature.

Given the extent of revision needed, we cannot make a decision about publication until we have seen the revised manuscript and your response to the reviewers' comments. Your revised manuscript is likely to be sent for further evaluation by all or a subset of the reviewers.

We expect to receive your revised manuscript within 3 months. However, if you require more time, please feel free to reach out to me.

**IMPORTANT - SUBMITTING YOUR REVISION**

*Re-submission Checklist*

*Published Peer Review*

*PLOS Data Policy*

*Blot and Gel Data Policy*

Sincerely,

Melissa

Melissa Vazquez Hernandez, Ph.D.

Associate Editor

PLOS Biology

ACADEMIC EDITOR'S COMMENTS:

I think R3 actually nails the key issues with her/his first two points. (1) R3 notes that “There is no experiment supporting the proposed role of DnaJ in vitro.” So I think the suggestion is that the authors should consider in vitro studies with DnaK/DnaJ/GrpE + bNACHT25. This is a pretty big ask, but because K/J/E has been studied extensively, this isn’t unthinkable either. I could see suggesting this, but indicating that we realize it could be a significant undertaking. (2) R3 also rightly points out (as R4 does too) that the authors experiment with the hyperactive bNACHT25 is over/misinterpreted as indicating that the folding of bNACHT25 is independent of DnaJ – it could be that the mutant makes bNACHT25 active enough that the role for DnaJ in promoting folding of the wild-type protein is obscured. R3 writes “The authors did not demonstrate whether bNACHT25 is soluble or insoluble in the absence of DnaJ. The use of the hyperactive stimulus-independent mutant does not prove that the folding of bNACHT25 WT is independent of DnaJ/K/E. In addition, it is not clear whether the WT and the hyperactive mutant are comparably expressed.” I agree that the authors need to assess levels of WT and mutant bNACHT25. And they need to assess whether bNACHT25 is soluble or insoluble without DnaJ.

I will add that even if the authors address point #2 above, and possibly/ideally point #1, they still need to adjust the wording and language of the paper. For instance, the language in the abstract and within the body of the paper about DnaJ “guarding” a host cell process is dangerously speculative and unfounded given the data actually presented. And the title implies that DnaJ is involved in phage detection which, I agree with R4 here, is misleading given that DnaJ may simply promote folding of bNACHT25 (either globally or even some domain of it) – seems a stretch to say that such a function implicates DnaJ in sensing per se.

REVIEWERS’ COMMENTS

— — —

Reviewer #2

In this revision, Conte, et al have added important and clever experiments that further support a role for DnaJ in bNACHT detection. Among other experiment, they use a hyperactive NACHT25 mutant that requires no stimulus and show that bNACHT is still active in the absence of DnaJ, indicating a role for DnaJ in sensing rather than bNACHT folding. This reviewer believes that the new data makes the work compelling and rigorous enough to be published in PLoS Biology. However, they should revise their language throughout the article to reflect the fact that the role of DnaJ in sensing (rather than degradation) is only clear when they show Figure 5. Likewise, it still remains unclear if DnaJ has a direct or indirect role in sensing. Given its basic function, and how little is still known about the sensing mechanism, it is still possible that DnaJ is simply involved in folding other factors (phage proteins or another sensing component) that enable detection. Their proposed model that bNACHT detects CP proteins through its interaction with DnaJ is exciting and compelling, but they should acknowledge the possibility that DnaJ might play an indirect role in the results and discussion.

— — —

Reviewer #3

Comments to the authors:

The role of DnaJ and it chaperone/cochaperone partners is phage detection is very interesting.

The authors have made significant changes and added several experiments to further address the role of DnaJ in vivo. However, the mechanism by which DnaJ mediates bNACHT activation by phage proteins is still not provided. As it is, we are left with an interesting in vivo observation but no real mechanistic breakthrough.

-There is no experiment supporting the proposed role of DnaJ in vitro.

-The authors did not demonstrate whether bNACHT25 is soluble or insoluble in the absence of DnaJ. The use of the hyperactive stimulus-independent mutant does not prove that the folding of bNACHT25 WT is independent of DnaJ/K/E. In addition, it is not clear whether the WT and the hyperactive mutant are comparably expressed.

-The fact that the DnaK/J/E chaperone machine is involved in this process, and not only DnaJ, is interesting but as for DnaJ, the mechanism of sensing is not provided (i.e., does it mediate bNACHT folding, sequestration or degradation in the absence of phage, do phage proteins compete with bNACHT for binding to DnaJ, etc..).

— — —

Reviewer #4

In this revised manuscript, the authors have added some experiments and addressed some of the reviewers' comments. However, they have not solved the fundamental problem with the paper. The novelty that the authors are trying to push here is that bNACHT25/32 is "guarding a host cell process rather than binding a specific phage-derived molecule." The authors have still provided no direct proof for this idea. Even the indirect proof is weak.

In their abstract, the authors claim:

1. "Here we determine the mechanism of RNA phage detection by the bacterial NLR-related protein bNACHT25 in E. coli." This is not true. The escaper mutants suggest a role for CP. However, the authors could not demonstrate a direct interaction between bNACHT and CP. CP and bNACHT co-expression from plasmids induced DNA degradation and cell death. However, the plasmid assay is non-physiological because both proteins are produced at much higher levels than in the experiments testing phage replication. The authors have not shown that the bNACHT system kills cells during phage infection, or that the DNA degradation observed in the plasmid assay is the mechanism for blocking phage replication. They have not shown that the detection of DNA degradation activity (i.e. 40 min after infection for bNACHT32) correlates with the MS2 infection cycle (i.e. how long after phage infection are infectious phage particles produced). So, we do not know how bNACHT is detecting phage infection and we do not know how it is blocking phage infection.

2. "A genetic assay confirmed CP was sufficient to activate bNACHT25 but the two proteins did not directly interact." CP is not proven to be sufficient for activation during phage infection. On the other hand, the non-detection of a CP:bNACHT interaction using co-IP is not conclusive proof of anything. A strong transient interaction with CP (i.e. fast on- and off-rates) or one that is too weak to detect by co-IP may be sufficient to activate bNACHT.

3. "Instead, we found bNACHT25 requires the host chaperone DnaJ to detect CP." The authors have shown that DnaJ is required for functioning of the bNACHT system. They have not in any way shown that DnaJ is required to detect CP. For this conclusion, the authors are relying on a new result: "We now provide a crucial experiment to demonstrate that the bNACHT25 protein is fully functional in dnaJ-deficient cells (Fig 5D and 5E). We have used an inducible system to express a hyperactive allele of bNACHT25. In these experiments, bNACHT25 is active in a stimulus-independent manner and disruption of dnaJ did not impact nuclease activity (Fig 5D and 5E). These data complement data already found in the manuscript showing that bNACHT25 is similarly expressed in wild-type and dnaJ-mutant strains (Fig 4C)." This experiment is not convincing. First, there is actually less DNA degradation by the H506L mutants in the dnaJ-deficient strain, so it is not fully functional. Second, this assay is non-physiological, and measures only the DNA degradation activity of NACHT25. It does not determine whether it can still bind to another protein and be activated.

4. "Our data suggest that bNACHT25 detects a wide range of phages by guarding a host cell process rather than binding a specific phage-derived molecule." There is no evidence at all that NACHT25 guards a host cell process involving DnaJ. A fundamental problem with this statement is that the authors provide no data to show that MS2 infection has any impact on DnaJ. In my previous review, I suggested that MS2 infection may increase DnaJ expression. However, the authors provided new data to show that this is not true. In the discussion, the authors push their idea further saying, "bNACHT25 adopts a mechanism of sensing most similar to the guard model and senses perturbations to the host cell proteome. CP is not a conventional pathogen effector; however, it still impacts a core host process that bNACHT25 monitors as a proxy for infection." This speculation goes way beyond any data presented here since the authors have not identified any change in a DnaJ-related process that might be detected. They have not even shown that CP binds to DnaJ, which might support this idea.

The most important finding in this paper, as described in the title, is that DnaJ mediates the activity of these bNACHT proteins. Given that DnaJ could be affecting bNACHT activity through many mechanisms (most likely by stabilizing or helping to form its active conformation), the onus is on the authors to provide strong evidence for their conclusion, which would be very exciting if true. However, the only evidence they have for this conclusion is their inability to detect a direct interaction between NACHT and CP, and their flawed experiment showing that bNACHT is still active in the absence of DnaJ. This is not enough. Thus, I feel strongly that publishing this manuscript in its current form and with its current emphasis would be very misleading. I am not saying that the authors' hypothesis is wrong, but publishing it will require insight into the mechanism. The pathway way to this insight will likely require a greater understanding of how bNACHT and DnaJ affect the MS2 lifecycle (i.e. do more phage infection experiments). Given that this may entail considerably more experiments, the authors might consider publishing the current data, but with a different emphasis.

---

## [Editor Report · Decision Letter 2]

22 Apr 2025

Dear Aaron,

Thank you for your patience while we considered your revised manuscript "Phage detection by a bacterial NLR-related protein is mediated by DnaJ" for publication as a Discovery Report at PLOS Biology. This revised version of your manuscript has been evaluated by the PLOS Biology editors and the Academic Editor.

Based on our Academic Editor's assessment of your revision, we are likely to accept this manuscript for publication, provided you satisfactorily address the remaining editorial points. Please also make sure to address the following data and other policy-related requests.

a) We routinely suggest changes to titles to ensure maximum accessibility for a broad, non-specialist readership, and to ensure they reflect the contents of the paper. In this case, we would suggest a minor edit to the title, as follows. Please ensure you change both the manuscript file and the online submission system, as they need to match for final acceptance:

"Phage detection by bacterial NLR-related bNACHT25 is mediated by DnaJ"

b) The maximum number of figures for a Discovery Report is 4. Currently you have 5 main figures. Please reduce them to 4 by either combining them or sending some to the supplementary material.

Please supply the numerical values either in the a supplementary file or as a permanent DOI’d deposition for the following figures:

Figure 1BCD, 2C, 3AD, 4ADE, 5DE, S2A-D, S5B, S6

d) Please cite the location of the data clearly in all relevant main and supplementary Figure legends, e.g. “The data underlying this Figure can be found in S1 Data” or “The data underlying this Figure can be found in https://doi.org/10.5281/zenodo.XXXXX”

e) Please ensure that your Data Statement in the submission system accurately describes where your data can be found and is in final format, as it will be published as written there.

f) Per journal policy, if you have generated any custom code during the course of this investigation, please make it available without restrictions upon publication. Please ensure that the code is sufficiently well documented and reusable, and that your Data Statement in the Editorial Manager submission system accurately describes where your code can be found.

We expect to receive your revised manuscript within two weeks.

*Published Peer Review History*

*Press*

Sincerely,

Melissa

Melissa Vazquez Hernandez, Ph.D.

Associate Editor

PLOS Biology

---

## [Editor Report · Decision Letter 3]

6 May 2025

Dear Aaron,

I hope you are doing great. Thank you for the submission of your revised Discovery Report "DnaJ mediates phage sensing by the bacterial NLR-related protein bNACHT25" for publication in PLOS Biology. On behalf of my colleagues and the Academic Editor, Michael T. Laub, I am pleased to say that we can in principle accept your manuscript for publication, provided you address any remaining formatting and reporting issues. These will be detailed in an email you should receive within 2-3 business days from our colleagues in the journal operations team; no action is required from you until then. Please note that we will not be able to formally accept your manuscript and schedule it for publication until you have completed any requested changes.

PRESS

Sincerely, 

Melissa

Melissa Vazquez Hernandez, Ph.D., Ph.D.

Associate Editor

PLOS Biology
